# Protective Role of an Extract Waste Product from *Citrus bergamia* in an In Vitro Model of Neurodegeneration

**DOI:** 10.3390/plants12112126

**Published:** 2023-05-27

**Authors:** Jessica Maiuolo, Francesca Bosco, Lorenza Guarnieri, Saverio Nucera, Stefano Ruga, Francesca Oppedisano, Luigi Tucci, Carolina Muscoli, Ernesto Palma, Angelo Maria Giuffrè, Vincenzo Mollace

**Affiliations:** 1Laboratory of Pharmaceutical Biology, IRC-FSH Center, Department of Health Sciences, School of Pharmacy and Nutraceutical, Faculty of Pharmacy, University “Magna Græcia” of Catanzaro, 88100 Catanzaro, Italy; 2IRC-FSH Center, Department of Health Sciences, Faculty of Pharmacy, University “Magna Græcia” of Catanzaro, 88100 Catanzaro, Italy; fra.bosco88@gmail.com (F.B.); lorenzacz808@gmail.com (L.G.); saverio.nucera@hotmail.it (S.N.); rugast1@gmail.com (S.R.); oppedisanof@libero.it (F.O.); l.tucci@head-sa.com (L.T.); muscoli@unicz.it (C.M.); palma@unicz.it (E.P.); mollace@libero.it (V.M.); 3Department of Agraria, University of Studies “Mediterranea” of Reggio Calabria, 89124 Reggio Calabria, Italy; amgiuffre@unirc.it; 4Faculty of Pharmacy, San Raffaele University, 00042, Rome, Italy

**Keywords:** *Citrus bergamia*, fiber of bergamot, byproducts, pastazzo, β-amyloid protein, Alzheimer’s disease, neurons, oligodendrocytes

## Abstract

A balanced diet, rich in fruits and vegetables and ensuring the intake of natural products, has been shown to reduce or prevent the occurrence of many chronic diseases. However, the choice to consume large quantities of fruits and vegetables leads to an increase in the amount of waste, which can cause an alteration in environmental sustainability. To date, the concept of a “byproduct” has evolved, now being understood as a waste product from which it is still possible obtain useful compounds. Byproducts in the agricultural sector are a rich source of bioactive compounds, capable of possessing a second life, decreasing the amount of waste products, the disposal costs, and environmental pollution. A promising and well-known citrus of the Mediterranean diet is the bergamot (*Citrus bergamia*, Risso et Poiteau). The composition of bergamot is known, and the rich presence of phenolic compounds and essential oils has justified the countless beneficial properties found, including anti-inflammatory, antioxidant, anti-cholesterolemic, and protective activity for the immune system, heart failure, and coronary heart diseases. The industrial processing of bergamot fruits leads to the formation of bergamot juice and bergamot oil. The solid residues, referred to as “pastazzo”, are normally used as feed for livestock or pectin production. The fiber of bergamot (BF) can be obtained from pastazzo and could exert an interesting effect thanks to its content of polyphenols. The aims of this work were twofold: (a) to have more information (composition, polyphenol and flavonoid content, antioxidant activity, etc.) on BF powder and (b) to verify the effects of BF on an in vitro model of neurotoxicity induced by treatment with amyloid beta protein (Aβ). In particular, a study of cell lines was carried out on both neurons and oligodendrocytes, to measure the involvement of the glia and compare it with that of the neurons. The results obtained showed that BF powder contains polyphenols and flavonoids and that it is able to exercise an antioxidant property. Moreover, BF exerts a protective action on the damage induced by treatment with Aβ, and this defense is found in experiments on the cell viability, on the accumulation of reactive oxygen species, on the involvement of the expression of caspase-3, and on necrotic or apoptotic death. In all these results, oligodendrocytes were always more sensitive and fragile than neurons. Further experiments are needed, and if this trend is confirmed, BF could be used in AD; at the same time, it could help to avoid the accumulation of waste products.

## 1. Introduction

In the last decades, the use of natural products has been increasingly promising and interesting, since they have been proven to possess numerous beneficial properties for human health [1,2,3]. In particular, a balanced diet, rich in fruits and vegetables and ensuring the intake of natural products, has been shown to reduce or prevent the occurrence of many chronic diseases [4]. However, the consumption of large amounts of fruit and vegetables has led to an increase in waste products. Citrus fruits are one of the most produced crops in the world, and data provided by the Food and Agriculture Organization Corporate Statistical Database (FAOSTAT) indicate that their world production was 158,490,986 tons in 2020. Citrus waste products are estimated to be around 15 million tonnes per year, worldwide [5]. Since these waste products are made up of a high content of bioactive compounds, they should be eliminated in a responsible and eco-friendly way, thereby increasing disposal costs [6,7]. As a result, (a) alteration of environmental sustainability and (b) an increase in environmental pollution may also occur. Currently, in order to minimize the growing production of waste generated in the agricultural sector, new regulations are being drawn up to define criteria for food waste management. In this direction, the concept of a “byproduct” has evolved, being a mixture of compounds from which something can still be obtained [8]. Byproducts represent a rich source of bioactive compounds, and in recent decades, many alternative methods of using fruit and vegetable waste have been developed [9,10,11]. Citrus byproducts can be divided into skins (flavedo and albedo), seeds and pulp residue. These byproducts possess several compounds with powerful bioactive activities for human health, with therapeutical effects on cancer, high blood pressure, diabetes, obesity, and neurodegenerative diseases [12]. In Italy, the main agricultural production includes the processing of citrus fruits and the oil industry. In particular, citrus fruits are not only widely grown and consumed, but also processed in the form of juices, concentrates, jams, canned fruit, dehydrated products, flavoring agents, beverages, and so on. This use of citrus is justified by its pleasant taste and numerous beneficial properties, including antioxidant, anti-inflammatory, anti-infective, anti-cancer, and neuroprotective effects [13,14,15]. A promising and well-known citrus of the Mediterranean diet is the bergamot (*Citrus bergamia*, Risso et Poiteau). This fruit belongs to the *Rutaceae* family and to the genus *Citrus* and, despite being present in some other areas of the world (Greece, Antilles, Canary Islands), preferably grows in a thin strip of coast in the province of Reggio Calabria, Calabria, Italy [16,17]. The reasons for the optimal growth of bergamot in this area are to be found in the climate and soil composition, which are optimal for the needs of the plant [18]. The intense scent of bergamot has made this fruit one of the main components in the production of perfumes, cosmetics, food, and sweets. In addition, popular medicine has used this citrus against fever and numerous microbial infections of the mouth, skin, and respiratory and urinary tracts [19,20]. The composition of bergamot is known, and the rich presence of phenolic compounds and essential oils has justified the countless beneficial properties found, including anti-inflammatory, antioxidant [21,22], anti-cholesterolemic, and protective activity for the immune system, heart failure, and coronary heart disease [23,24]. Bergamot polyphenols are many, and among them, the most represented are naringin, neoeriocitrine, neohesperidin, and glycosylated polyphenols, such as bruteridine and melitidine. The polyphenolic fraction of bergamot (BPF) is obtained from citrus fruit (both albedo and flavedo) and is the fraction in which polyphenols are concentrated, reaching a total concentration of 40% [25]. Recent studies have shown that bergamot polyphenols exert not only an antioxidant response both in vitro and in vivo [26,27,28,29], but also a reduction in cholesterol, glucose, serum triglycerides, and systemic inflammation, and improvement of endothelial function [30,31,32,33,34,35]. The industrial processing of bergamot fruits leads to the formation of bergamot juice and bergamot oil. The solid residues consist of 50–55% peel, 10–15% pulp fragments, and 3–5% seeds. The set of these residues is referred to as “pastazzo”, and it is normally used as feed for livestock or pectin production [36]. With the further processing of the pastazzo, it is possible to obtain flavonoids, fats, amino acids, pectins, minerals, and fibers [37]. The fiber recovered from bergamot citrus (BF) consists of cellulose, hemicellulose, pectin, and inulin and, like most dietary fibers, is indigestible and not absorbable in the small intestine of humans [38]. This fraction, obtained from the pastazzo of bergamot, exerts an interesting effect thanks to its content of polyphenols.

Alzheimer’s disease (AD) is the most common form of neurodegenerative dementia that affects the elderly population. Pathologically, it is characterized by senile plaques and neurofibrillary tangles that occur in the brain area and produce neuronal loss. β-amyloid protein (Aβ) is a primary component of senile plaques and plays a primary role in the molecular pathology of AD [39]. In particular, Aβ is a hydrophobic peptide of 40–43 amino acids that derives from the cleavage of the precursor protein of the amyloid transmembrane (APP). Once formed, Aβ is highly prone to aggregation and has neurotoxic properties, facilitating neuroinflammatory processes and cortical thinning, reducing brain volume, and promoting neuronal death [40,41]. The balance between synthesis and clearance of this peptide is important to maintain its normal levels in the brain [42]. Recent contributions have highlighted the collaborative role of glia in AD: oligodendrocytes and astrocytes are, in fact, essential to ensure typical neuronal changes, such as reduction of synaptic density, alteration of electrophysiological properties, and neuronal degeneration [43]. It is known that diet is able to modulate neuroinflammatory processes in animals [44]; for example, bioactive molecules such as polyphenols, unsaturated fats, and antioxidant vitamins inhibit oxidative stress and neuroinflammation [45,46,47]. Brain studies have shown that some polyphenols belonging to the Mediterranean diet can reduce cognitive decline and the onset of AD [48,49]. In addition, these results have recently been supported by neuroimaging studies, which have shown protective effects of these compounds on neuronal structures and early morphological changes related to neurodegeneration and AD [50,51,52]. The purpose of this work is to test the effects of BF on an in vitro model of neurotoxicity induced by treatment with the Aβ protein. In particular, this model must include both neurons and oligodendrocytes so that we can measure the involvement of the glia and compare it to that of the neurons. For this reason, we will organize our work in a co-culture system, consisting of neurons and oligodendrocytes exposed to Aβ in a common growth medium, without ever coming into contact with each other. In addition, we will study the effects generated by pre-exposure to BF. If the effects of BF are encouraging, this could suggest a potential use of solid residues of bergamot in AD, assessing its effects in neurodegenerative diseases and, at the same time, helping to avoid the accumulation of waste products.

## 2. Materials and Methods

### 2.1. Plant Material

BF was obtained from the plant *Citrus bergamia* (Risso et Poiteau), from the harvest of February 2022 at Bianco, a small town in the province of Reggio Calabria, Italy. The fruit was properly peeled and squeezed to obtain three fractions:(1)bergamot juice;(2)bergamot oil;(3)pastazzo.

The pastazzo was ground and washed, and in the latter phase a water:pastazzo ratio of 2:1 was maintained. In the next phase, the suspension was subjected to mechanical separation (centrifugation at 4000 rpm for 10 min), generating a liquid phase and a solid phase. The solid phase was then dried (hot air at 60 °C) to a moisture content of less than 10%. The dried mash was ground to obtain a powder with an average particle size of 60 meshes, and this operation created the bergamot fiber.

### 2.2. HPLC Analysis

A portion of BF was subjected to chromatographic analysis (HPLC), and the data generated by this method provided interesting information on the extract. HPLC analysis was performed on a Perkin Elmer Flexar module equipped with a photodiode detector (PDA), a 200 series autosampler, a 200 series Peltier LC column furnace, a 200 series LC pump, and an Agilent 4 μm C18 100A column (250, 4.6 mm). HPLC system control and data collection were performed online by a computer equipped with Chromera software (v3.4.0.5712). A total of 1 g of BF dry powder was added to 20.0 mL of a 50:50 (*v*/*v*) water: ethanol solution to extract the polyphenolic fraction. The mixture was stirred for 3 h at 50 °C. An aliquot of the mixture was then filtered through a 0.2 μm PTFE filter and subjected to HPLC analysis. For elution, a two-solvent gradient (0.88% trifluoroacetic acid/acetonitrile) with a flow of 1.1 mL/min was used, maintaining the column at 30 °C. The detector wavelength was set to 284 nm. The HPLC system was calibrated with Naringin quantitative standard (purity > 99%) in a range of concentration of 10−1000 ppm. The flavonoids, as objects of interest, were quantified as naringin equivalents.

### 2.3. Oxygen Radical Absorbance Capacity (ORAC)

The antioxidant activity of BF was determined by the oxygen radical absorbance capacity (ORAC) assay, which is a method that measures the antioxidant activity of a sample by evaluating the transfer of a hydrogen atom. In particular, the fluorescence loss of fluorescein (used as probe) is measured over time. This fluorescence is due to the formation of peroxylic radicals, following the spontaneous degradation of 2,2′-azobis-2-methyl-propanimidamide, dihydrochloride (AAPH), which occurs at 37 °C. The peroxylic radical oxidizes the fluorescein, causing the gradual loss of the fluorescence signal. Antioxidants suppress this reaction and inhibit the loss of signal. In addition, 6-Hydroxy-2,5,7,8-tetramTethylchroman-2-carboxylic acid (Trolox) is a water-soluble analogue of vitamin E that inhibits the decay of fluorescence in a dose-dependent manner. Fluorescein and AAPH solutions were prepared in PBS (pH = 7.0) at concentrations of 0.02 mg/mL and 59.8 mg/mL, respectively. In contrast, Trolox was made in PBS (pH = 7.0) at concentrations of 7.65, 15.25, 30.5, and 61 μg/mL. Finally, BF was used at a concentration of 10 μg/mL. The evaluation of the fluorescent decay for fluorescein was conducted using a microplate reader, where excitation and emission wavelengths of 485 and 520 nm were used, respectively, at 37 °C. Measurements were carried out in triplicate every 2 min for 1.5 h, and the data obtained from the fluorescence vs. time curves are reported as the average antioxidant efficacy of the antioxidant compound. A regression equation was constructed by comparing the net area under the decay curve of the fluorescein and the Trolox concentration. The area under the curve was calculated with the following equation:i = 90
AUC = 1 + Σ f1/f0
i = 1

### 2.4. Measurement of Total Polyphenols by the Folin–Ciocalteu Assay

The total phenolic content of the BF was estimated by the Folin–Ciocalteu method, using naringin 99% (purchased by Sigma-Aldrich, St. Louis, MO, USA) as the reference standard for plotting the calibration curve. An extract of BF was prepared by adding 20 mL of ethanol/water 50:50 (*w*/*w*) to 1 g of powder to obtain absorbance values within the linearity range of the calibration curve. The mixture was stirred for 24 h at 20 °C. Cuvettes were prepared such that there were three replicates, and 400 μL of extract was added to 0.8 mL of 10-fold-diluted Folin–Ciocalteu reagent and shaken thoroughly. After 3 min, 0.8 mL of sodium carbonate 7% (*w*/*v*) was added, and the mixture was allowed to stand for 2 h with intermittent stirring until the color developed. The absorbance of the resulting blue colour was measured at 650 nm with a Prisma V-1200 Spectrophotometer. The total phenolic content was determined from the linear equation of a standard curve prepared with different concentrations of naringin, and the results were expressed as mg of naringin equivalent per g of dry weight.

### 2.5. Measurement of Flavonoids Content

The flavonoid content of the extract was measured by the colorimetric technique of aluminum chloride. Specifically, 1 mL of extract was mixed with 1 mL of 2% aluminum chloride in methanol. After 30 min, the absorbance at 430 nm was measured, and the equivalent of quercetin per gram of extract (mg QE/g extract) was used to represent the estimated content of flavonoids. This method is based on the findings of Kosalec et al. [53].

### 2.6. Antioxidant Activity through the DPPH Assay

The antioxidant activity of BF was measured using the stable radical 2,2′-diphenyl-1 picrylhydrazyl (DPPH) at a concentration of 4 mg/100 mL as reagent. In particular, the reduction in absorbance, visible as a change in color from purple to yellow, was measured when this radical reacted with a proton-donor antioxidant. For the experiments, 850 µL of DPPH solution was added to 50 µL of various extract concentrations (0.01–0.04), and a time of 20 min elapsed while the mixture was kept in the dark. Subsequently, absorbance measurements were made at 517 nm at room temperature in a UV–Vis spectrophotometer (Multiskan GO, Thermo Scientific, Denver, CO, USA). The results obtained were expressed as inhibition value % and IC50. The latter represents the concentration of the extract needed to remove 50% of the DPPH radicals. In order to obtain statistical significance, the test was performed three times.

### 2.7. Cell Cultures

Human cell line neurons (SH-SY5Y) and oligodendrocytes (MO3.13) were acquired from the American Type Culture Collection (Sesto San Giovanni, Milan, Italy), kept in culture in Dulbecco’s modified Eagle’s medium (DMEM) (enriched with 100 U/mL penicillin, 100 µg/mL streptomycin, 10% heat-inactivated fetal bovine serum), and grown in a humidified 5% CO_2_ atmosphere at 37% C. Cell lines have been grown in co-culture, using specific 12-well Transwell insert plates that allowed us to plate the two cell lines separated by a polyester membrane with 1 µm pores, to ensure close proximity of cellular lines while preventing cell migration. For this purpose, oligodendrocytes and neurons were placed on the bottom of the plate and the bottom of the Transwell inserts, respectively, and under these experimental conditions, only the growth medium came into contact with the cell lines. The model used in this experimental work is represented in Appendix A. Neurons and oligodendrocytes were placed on the bottom of the plate and inserts, respectively, and the growth medium was in contact with both cell lines. A representation of this model is shown in Appendix A. In order to achieve an optimal phenotypic appearance, oligodendrocytes and neurons were appropriately differentiated before treatment: MO3.13, a hybrid line resulting from a combination of adult human cells of rhabdomyosarcoma and oligodendrocytes, were differentiated with a treatment of Phorbol 12-myristate 13-acetate at 100 nm for five days [54]. For neurons (SH-SY5Y), a treatment using trans retinoic acid (Sigma Aldrich, Milan, Italy) at 10 µm for five days was chosen. The time chosen and the concentrations used to carry out the differentiation were drawn from the published scientific literature. The medium was changed every 2–3 days, and when the cell lines reached 50% confluence, they were treated with BF (10 μg/mL, dissolved in H_2_O) for 24 h or pretreated with BF for 24 h and then exposed to Aβ (20 μm) for 24 h. At the end of treatment, all appropriate tests were performed.

### 2.8. Proliferation Assay and Cytotoxicity Study

The colorimetric assay using 3-(4,5-dimethyl-2-yl)-2,5-diphenyltetrazole bromide (MTT) is capable of evaluating cell proliferation. In general, live cells have mitochondria with active enzymes, which can reduce MTT, resulting in colorimetric variation. Therefore, the appropriate measurement of MTT reduction provides information on cell viability and metabolic activity. For this purpose, 8 × 10^3^ cells/well were plated in 96-well plates. After 24 h, the growth medium was replaced with fresh medium containing BF 10 µg/mL or Aβ 20 µM, as described above. Subsequently, the medium was replaced with a red phenol-free medium containing an MTT solution (0.5 mg/mL) and, after 4 h incubation, 100 μL 10% SDS was added to each well to solubilize the formazan crystals. The optical density was measured at wavelengths of 540 and 690 nm using a spectrophotometer reader (X MARK Spectrophotometer Microplate Bio-Rad). Cell mortality was measured using the Trypan blue exclusion assay. By this test, it is possible to distinguish between viable cells, which possess intact cell membranes excluding the Trypan blue dye, and dead cells, which are characterized by damaged plasma membranes. They will let in the dye and, for this reason, will appear in blue. The cell suspension after treatment is mixed with the dye and examined under optical microscopy: the viable cells will be characterized by a clear cytoplasm, a consequence of the exclusion of the dye. Conversely, dead cells will appear blue due to dye absorption. A drop of the mixture, consisting of Trypan blue and cell suspension in equal measure, is applied in a hematocytometer, where stained (vital) and colored (non-viable) cells are counted under a microscope. Mortality is calculated as the ratio of the number of dead cells to the total number of cells × 100 [55].

### 2.9. Measurement of In Vitro Reactive Oxygen Species

H_2_DCF-DA is a molecule that easily spreads into cells and is cleaved to H_2_DCF, with loss of an acetate group, by intracellular esterases. H_2_DCF can no longer leave the cell and is oxidized by binding to intracellular reactive oxygen species (ROS) to form the highly fluorescent DCF compound. The quantification of the DCF probe provides the content of the ROS in the cell. Both cell lines were seeded in 96-well microplates with a density of 6 × 10^4^. The following day, they were treated with BF for 24 h. At the end of the treatment period, the growth medium was replaced by a fresh, phenol-free medium containing H_2_DCF-DA (25 μm). After exposure of 30 min at 37 °C, the cell lines were washed with PBS in order to remove excess H_2_DCF-DA, centrifuged, resuspended in PBS, and exposed or not to H_2_O_2_ (100 μM, 30 min), and fluorescence was evaluated by cytometric analysis (FACS Accury, Becton Dickinson, Franklin Lakes, NJ, USA).

### 2.10. Cell Lysis and Immunoblot Analysis

The total cell lysates of both lines were obtained from 6-well plates equipped with inserts. The cells were exposed to a preheated lysis buffer (80 °C) containing 50 mm of Tris-HCl (pH = 6.8), 2% of SDS, and a protease inhibitor mixture, and their lysates were immediately boiled for 2 min. The DCA assay determined the protein concentration of the extracts. We added 0.05% bromophenol blue, 10% glycerol, and 2% β mercaptoethanol. The samples were boiled again and loaded into SDS-polyacrylamide gels (12%). Subsequently, polypeptides were transferred to nitrocellulose filters, and blocked with TTBS/milk (TBS 1%, Tween 20, and non-fat dry milk 5%), and then antibodies were used to reveal the respective antigens. The primary antibodies were incubated overnight at 4 °C, followed by a secondary antibody conjugated with horseradish peroxidase for 1 h at room temperature. The stains were developed using the chemiluminescence procedure. The following primary antibodies were used: a polyclonal rabbit antibody for cleaved caspase-3 (Sigma-Aldrich, AB3623) at 1:1000 dilution, and a monoclonal mouse anti-actin antibody (Sigma Aldrich) at 1:5000 dilution. Horseradish peroxidase conjugated anti-mouse/anti-coniglio antibodies were used as secondary antibodies at 1:10,000 dilution.

### 2.11. Annexin V Staining

To assess the type of cell death, both cell lines were tripsinized, washed with cold PBS, and resuspended in a buffer containing annexin V/kit at a concentration of 1 × 106 cells/mL. First, 100 µL of the suspension was transferred in a new tube; then, 5 mL of FITC annexin V (BD Biosciences, San Jose, CA, USA) was added. After incubation of samples for 15 min in the dark, 400 µL of 1× binding buffer and 5 µL of propidium iodide (PI) were added to each tube and incubated for 1 h. A total of 30,000 cells per sample were acquired by flow cytometry (emission filter 515–545 nm for FITC; 600 nm for PI) using a cytofluorometer (FACS Accuri, Becton Dickinson, Milan, Italy).

### 2.12. Immunofluorescence

Immunofluorescence is an immunology technique coupled to the fluorescence phenomenon that allows one to observe and identify specific proteins by means of specific antibodies that emit a light signal. The cell lines were plated (1 × 10^6^) in specific 6-well Transwell insert plates on slides. The cells have been treated as described and appropriately fixed. After PBS washing (three times, five min), non-specific cell sites were blocked with 5% BSA in PBS. The cells were incubated with a primary antibody (mouse monoclonal anti-beta amyloid antibody, ab11132, abcam, at dilution 1:200) and diluted in BSA 2.5% overnight at 4°, under constant agitation. The following day, after washing the sample with PBS (3 times, 5 min), the slides were exposed to secondary mouse antibody (conjugate with Alexa Fluor 488, Sigma Aldrich, at diluition 1:1000) and diluted in BSA 2.5% for 1 h at room temperature. After a further three washes, the slide was mounted properly using 80% glycerol or a commercial reagent as a pillar. Finally, the confocal microscope visualization was carried out.

## 3. Results

### 3.1. HPLC Analysis of BF

In order to check whether or not the polyphenols contained in BPF, due to which it is possible to justify the antioxidant action of this fruit, were present in the BF fraction, an HPLC analysis was performed following the extraction of BF. As can be observed from the spectrum shown in Figure 1 and from the quantitative report included at the bottom, our objects of interest, flavonoids (brutieridine, melitidine, naringine, neoeriocitrine, neohesperidine) are present.

### 3.2. Total Polyphenols and Flavonoids

In Table 1, BF is represented in terms of its total content of polyphenols and flavonoids. Folin–Ciocalteu reagent and aluminum trichloride allowed us to measure the content of polyphenols and flavonoids, respectively, in BF extract. From the different naringin concentrations, the regression equation for the polyphenol content (y = 0.2159x, R^2^ = 0.9867) gave 48.8 ± 5 mg E-naringin/g dry extract. The calibration curve for quercetin and the regression equation (y = 0.1821 + 0.1173, R^2^ = 0.991) show that the concentration of flavonoids equivalent to the concentration of quercetin of BF was equal to 6.21 ± 2.9 mg E-Q/g dry extract. The content of polyphenols is confirmed by the antioxidant capacity of the powder tested with the ORAC test and could justify the antioxidant effects obtained in our experimental model of neurons and oligodendrocytes.

### 3.3. DPPH Free Radical Scavenging Activity

The inhibitory concentration IC50 of BF was calculated after measuring the absorbance. BF has been shown to have a robust antioxidant capacity, lower than the BHT standard, but still high, with an IC50 of 0.0085 + 0.00016 mg/mL. In particular, the IC50 of the BHT standard was 0.0032 + 0.00016 mg/mL (Figure 2).

### 3.4. Antioxidant In Vitro Activity

This test was selected and optimized with a final concentration of 59.8 mg/mL for AAPH and 0.02 mg/mL for fluorescein. The prepared Trolox concentrations resulted in a linear relationship with the net area under the curve (AUC). The following step was to apply the sample to be tested (BF). In this test, the maintenance of the fluorescence signal is indicated as the area under the curve (AUC) measured over time. The relative value of BF was obtained by comparing its AUC to that of the standard antioxidant curves, generated by different concentrations of Trolox. The results showed that BF possesses an antioxidant capacity; in fact, its corresponding curve was positioned between the curves of two Trolox concentrations: 7.6 μg/mL and 15.25 μg/mL. The results of the ORAC assay, relative to BF, are reported in Figure 3.

### 3.5. Effects of BF and Aβ on Viability of Neurons and Oligodendrocytes

First, we tested different concentrations of BF on both cellular lines (0.1, 1, 10, 100, 200, 500, and 1000 μg/mL) in order to know if it could be toxic to cells. As can be seen in Figure 4a, no concentration of BF is toxic, and only the highest dose (1000 μg/mL) shows a statistically significant minimum of toxicity. In addition, no significant difference was found between SH-SY5Y and MO3.13. In these experiments, treatment with STAURO, an antibiotic isolated from the bacterium *Streptomyces staurosporeus* and capable of inducing apoptosis, was used as a positive control. Since most BF concentrations did not cause cellular damage, we chose to continue the experimental work using the intermediate dose of 10 μg/mL. The choice of this concentration was made with the intention of avoiding high concentrations of BF, which could create a potential synergistic effect in the co-treatment with BF-Aβ. In addition, this concentration is the only one among those tested that is able to significantly prevent damage induced by treatment with Aβ, as can be seen in Figure 5a. In order to know the effects generated by BF when administered for a longer time (48 and 72 h), specific experiments on cell viability have been conducted, as shown in Appendix A. Subsequently, we evaluated the effects of Aβ on both cell lines. The results of viability and mortality are represented in Figure 4c,d, respectively, following treatment with the neurotoxic peptide.

As can be seen, the toxicity induced by Aβ is concentration dependent in both cell lines. However, a difference is evident: in both the viability and mortality experiments, oligodendrocytes appear more sensitive than neurons, showing greater frailty with regard to the toxic damage induced by Aβ. Regarding the peptide Aβ, we have chosen a concentration of 20 μM, as it is responsible for a mortality between 40 and 60% and is adequate to try to reduce this toxic damage. These results are shown in Figure 4 below.

### 3.6. BF Significantly Protects against Damage Induced by Aβ

In order to assess the effect of BF on damage induced by Aβ, appropriate experiments were carried out, in which cell lines were treated with both BF and Aβ, as described. As can be seen in Figure 5a, of all BF concentrations used (1, 10, 50, 100, and 200 μg/mL), only a dose of 10 μg/mL is able to significantly protect both cell lines from damage induced by Aβ. In addition, we can appreciate a greater frailty of oligodendrocytes than neurons. Finally, the same result was achieved and evidenced by immunofluorescence experiments, in which the cells were incubated with a primary anti-beta-amyloid antibody. As can be observed in Figure 5b, cells treated with Aβ appear luminescent and bright, while pretreatment with BF 10 μg/mL turns off the fluorescence of both cell lines.

### 3.7. Antioxidant Role of BF

Aβ treatment causes the formation and accumulation of reactive oxygen species, detectable in both cell lines when these treatments are compared to untreated cells. In particular, the increase in ROS is represented by the shift to the right of the cell peak, as indicated by the corresponding percentage. These results are represented in Figure 6a. BF alone (at different concentrations of 1, 10, and 50 μg/mL) does not possess an oxidative effect, and their peaks are similar to those of the the untreated cells (data not shown). The effects of co-treatment (BF + Aβ) show that only the dose of 10 μg/mL causes a significant reduction of ROS, as indicated by the shift to the left of the fluorescence peak. On the contrary, concentrations of 1 μg/mL and 50 μg/mL are ineffective. Despite the similarity of the response, once again oligodendrocytes prove to be more delicate than neurons, maintaining a greater amount of ROS. Finally, H_2_O_2_ treatment (100 μM, 30 min) is used as a positive control to demonstrate the functioning of the system. In Figure 6b, the respective quantification is represented.

### 3.8. BF Protects against Apoptotic Death Induced by Aβ

In order to know the type of death generated by Aβ, we studied the expression of caspase-3, which is generally involved in the process of apoptotic death, in our experimental model. As shown in Figure 7a, Aβ treatment was able to increase the expression of cleaved caspase-3 compared to the control; BF alone caused no or little damage, while co-treatment with BF + Aβ generated a significant reduction in the expression of caspase-3, which appeared to be less than the treatment with Aβ but greater than the control. In Figure 7b, the respective quantification of these results is reported. In Figure 7c, an annexin V-PI staining experiment is highlighted, in which we were able to discriminate between types of cell death. As can be seen, the control is represented only by viable cells; Aβ induces the displacement of most cells in the box of late apoptosis, while co-treatment BF + Aβ protected cells, and a large amount remains viable. In these experiments, it was particularly noticeable that BF resulted in more protection in neurons than in oligodendrocytes, as can be inferred from the comparison of co-treated cells that remain viable: in fact, neurons are numerically more than oligodendrocytes. In Figure 7d, the respective quantification is reported.

## 4. Discussion and Conclusions

In our experiments, first of all, we wanted to test the effects of increasing concentrations of BF on both cell lines, in order to exclude the toxicity of each fraction. In fact, since treatment with Aβ causes a concentration-dependent increase in mortality, on both neurons and oligodendrocytes, we wanted to be sure that the damage detected was not also motivated by an additive effect generated by BF (Figure 4). In addition, the BF fraction was not only administered to the cells for 24 h, but also for longer times in order to ensure the non-toxicity of this fraction (Appendix A). These results, in line with the treatment conducted at 24 h, showed the same trend: the BF fraction was not substantially toxic, and the only reduction in cell viability (10–15%) was found at the higher concentrations (500 and 1000 μg/mL), which were not used in this experimental work. The protective effects exerted by BF against damage induced by Aβ are reported and well described in Figure 5. First of all, only BF 10 μg/mL was responsible for the significant reduction of mortality detected after treatment with Aβ alone. Moreover, immunofluorescence experiments have highlighted how BF was able to turn off the fluorescence signal that we measured after treatment with Aβ and the exposure of cells to beta amyloid antibody. The protective effects exerted by BF could be explained by its chemical composition; in fact, the HPLC analysis suggested the presence of many protective polyphenols such as brutieridine, melitidine, naringin, neoeriocitrine, and neohesperidine, among others (Figure 1). In recent years, numerous studies in the scientific literature have shown a protective action by these compounds against the neurodegenerative pathology of AD [56,57,58,59]. In addition, a quantification test of total polyphenols and flavonoids carried out on BF showed that their content was 48.8 ± 5 and 6.21 ± 2.9 mg dry extract, respectively (Table 1). The polyphenol content allegedly justifies the proven antioxidant role of the extract and is obtained both in dust (measured through DPPH free radical scavenging assay, Figure 2, and the ORAC test, Figure 3) and on cell lines of neurons and oligodendrocytes (measured by evaluation of the DCF fluorescence probe, Figure 6). This specific effect of BF is very important since Aβ causes the formation and accumulation of ROS, the common denominator of most neurodegenerative pathologies, including AD [60,61,62]. The reduction of cell viability, which is Aβ-induced, appears to be due to the activation of an apoptotic process, as confirmed by the scientific literature [63,64,65]. In fact, the treatment with Aβ has determined an increase in the expression of caspase-3, while the pre-treatment with BF has reduced it. Since it has been shown that activation of caspase-3 can sometimes occur independently of activation of apoptosis [66,67,68,69], we performed the annexin-PI assay that provided information on the type of cell death induced. This test confirmed the progression of the Aβ-induced apoptosis and significant protection exerted by BF (Figure 7).

Citrus fruits are among the most consumed fruits in the world, but they are also one of the most significant sources of food waste considering the immense potential of bio-products included in waste products. Today, it is increasingly known that citrus waste has a huge industrial potential, being used to obtain polymers suitable for replacing plastic, to supply products for food packaging or coating, and to create food additives and stabilizers of nanoparticles [70]. For this reason, it is essential to develop new and more efficient methods of extraction, capable of promoting the production of bio-products by reducing the use of aggressive or toxic solvents and the amount of energy used, increasing yields without compromising the extracts [71]. In our experimental work, BF at 10 μg/mL has demonstrated a protective effect on the damage induced by Aβ, and if this effect was confirmed in further in vitro, in vivo, and clinical trials, it could be used for curative purposes against AD. In general, we can say that the products of bergamot waste, and in particular BF, could be used to generate bioactive compounds and have a second life, without increasing disposal costs and environmental pollution [72]. The second topic to be developed as a result of the data obtained is that oligodendrocytes were more fragile and susceptible than neurons in the present experimental model. In particular, Aβ has caused more damage in MO3.13 than in SH-SY5Y, and the protection of BF 10 μg/mL, exercised in equal measure on both cell lines, has failed to match the entity of defense in neurons and oligodendrocytes. This aspect is really very interesting; in fact, if it is further confirmed, it could indicate the glia as the weakest fraction compared to neurons. Moreover, if treatment with Aβ causes more damage to the glia in AD, the neurons would no longer be the main actors, but, as in a Copernican revolution, the glia could become the main pharmacological target. However, even in this case, further confirmation would be needed in order to turn the present speculations into concrete hypotheses.

## Figures and Tables

**Figure 1 plants-12-02126-f001:**
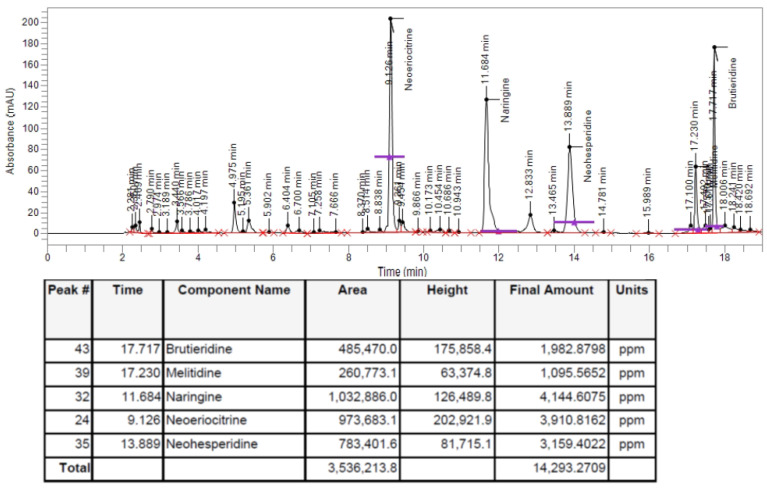
HPLC analysis of BF. Chromatogram of HPLC analysis of bergamot fiber from *Citrus bergamia*.

**Figure 2 plants-12-02126-f002:**
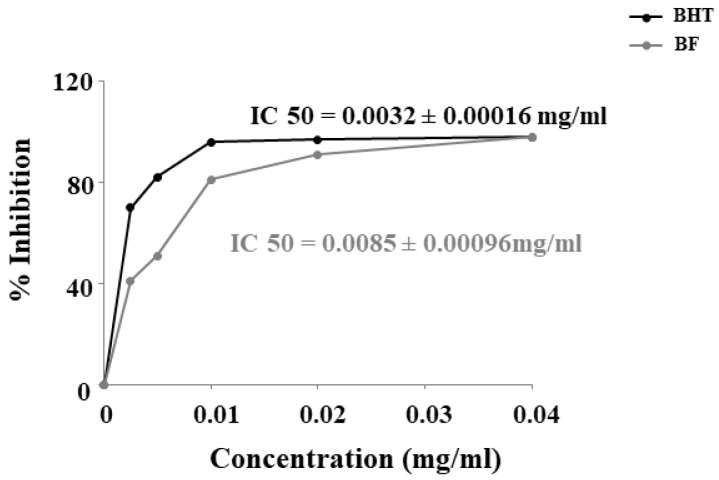
DPPH scavenging activity of BHT and BF.

**Figure 3 plants-12-02126-f003:**
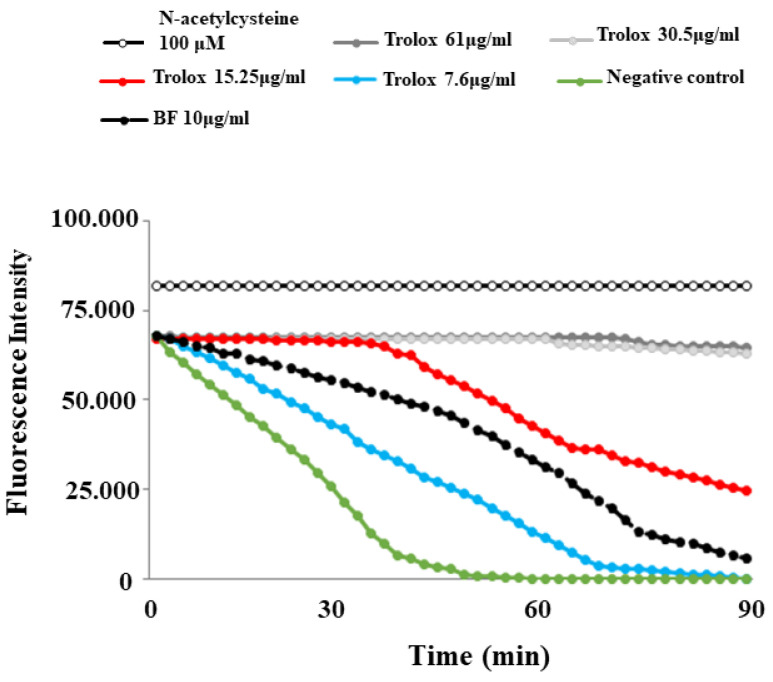
ORAC assay decay curves for Trolox and BF.

**Figure 4 plants-12-02126-f004:**
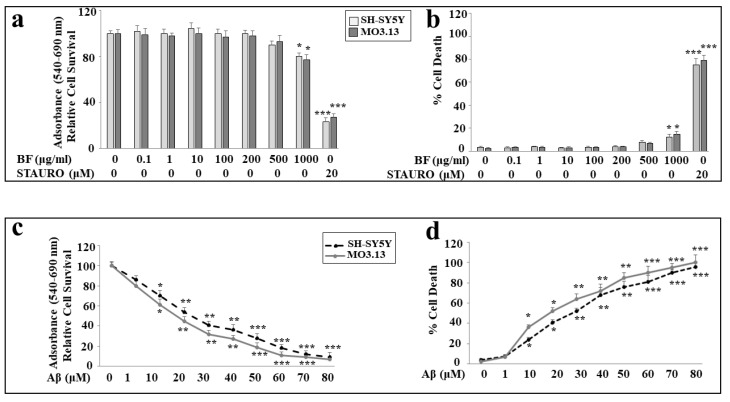
Effects of BF and Aβ on vitality of both cell lines. In panel (**a**), the representation of cell viability following increasing concentrations of BF on neurons and oligodendrocytes is shown. The experiments were conducted using MTT assay. Under the same experimental conditions, cell mortality is shown in panel (**b**) (Trypan blue exclusion assay). In these experiments, treatment with STAURO, an apoptosis inductor, was used as a positive control. Panels (**c**,**d**) show the toxic effects induced by increasing concentrations of Aβ on neurons and oligodendrocytes on cell viability (MTT assay) and cell mortality (Trypan blue exclusion assay), respectively. Three independent experiments were carried out, and the values are expressed as the mean ± standard deviation (sd). * denotes *p* < 0.05 vs. the control; ** denotes *p* < 0.01 vs. the control; *** denotes *p* < 0.001 vs. the control. Analysis of variance (ANOVA) was followed by a Tukey–Kramer comparison test.

**Figure 5 plants-12-02126-f005:**
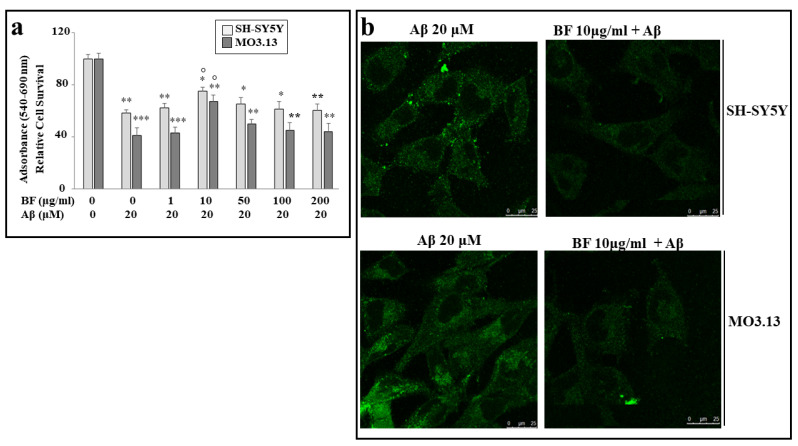
Protection by BF against damage induced by Aβ. In panel (**a**), only pre-treatment with BF 10 μg/mL is able to protect cells from Aβ-induced damage, and this result can be appreciated by observing the significant increase in viability compared with treatment with Aβ alone. All other BF concentrations tested showed no protective effects on neurons or oligodendrocytes. In panel (**b**), immunofluorescence images of SH-SY5Y and MO3.13 cells are reported. In particular, the cells were incubated with anti-beta-amyloid antibody. Above, each box indicates the specific treatment related to the image. In the panels (**a**,**b**), three independent experiments were carried out, and the values were expressed as mean ± sd. * denotes *p* < 0.05 vs. the control; ** denotes *p* < 0.01 vs. the control; *** denotes *p* < 0.001 vs. the control. ° denotes *p* < 0.05 vs. Aβ. Variance analysis (ANOVA) was followed by a Tukey–Kramer comparison test.

**Figure 6 plants-12-02126-f006:**
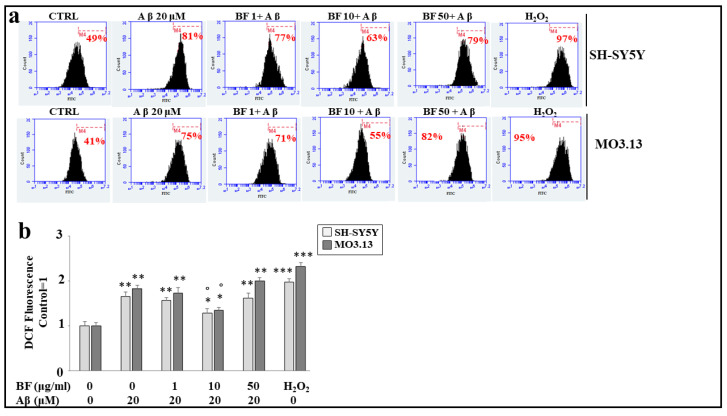
BF 10 μg/mL is able to protect both cell lines from oxidative damage produced by treatment with Aβ. Panel (**a**) represents the change of cellular fluorescence as a result of the treatment carried out. In particular, every single box is generated following the reading of the fluorescence of the cells: on the *x*-axis, the fluorescence is represented (among the fluorochromes, we have chosen FITC, fluorescein isothiocyanate, which binds to our fluorescent probe), while the *y*-axis is relative to the number of cells that we decided to acquire. At the top of each graph, there is a marker (M4), which is arbitrarily drawn in the control and kept the same for all other samples. The part of the peak included in the marker is indicated by a numerical percentage. In panel (**b**), the quantification is obtained and displayed as representations of the various percentages. The control percentages are arbitrarily made equal to 1, and the other values are related to it. Three independent experiments were performed, and the values were expressed as the mean ± sd. * denotes *p* < 0.05 vs. the control; ** denotes *p* < 0.01 vs. the control; *** denotes *p* < 0.001 vs. the control. ° denotes *p* < 0.05 vs. Aβ. Variance analysis (ANOVA) was followed by a Tukey–Kramer comparison test.

**Figure 7 plants-12-02126-f007:**
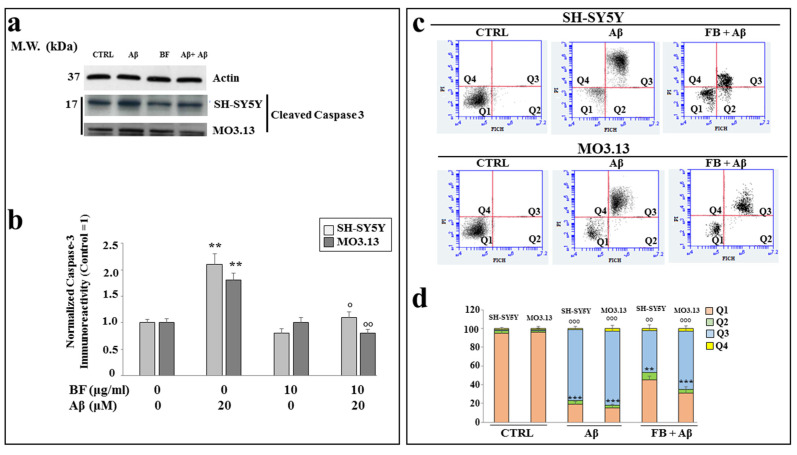
Results regarding the expression of caspase-3 and annexin V-PI staining. In panel (**a**), the expression of the cleaved fraction of caspase-3 is shown. The results have been normalized thanks to the housekeeping actin protein. In panel (**b**), the respective quantification is highlighted. Three independent experiments were carried out, and the values were expressed as the mean ± sd. ** denotes *p* < 0.01 vs. the control. ° denotes *p* < 0.05 vs. Aβ. °° denotes *p* < 0.01 vs. Aβ. Analysis of variance (ANOVA) was followed by a Tukey–Kramer comparison test. In panel (**c**), annexin V-PI staining is shown. The cytofluorometric analysis, conducted on each sample, generated these plots; in the *x*-axis, there is the fluorophore FITC that binds to annexin, and in the *y*-axis, the fluorophore PI that binds to propidium iodide. Each plot is divided into 4 quadrants: Q1, in which the cells are basically viable and are annexin-PI negative; Q2, in which the cells are annexin positive and PI negative, corresponding to cells undergoing an early apoptotic process; Q3, in which there are annexin-PI-positive cells, identifiable with dead cells through a process of late apoptosis; and Q4, which highlights annexin-negative cells and PI-positive cells, represented by dead cells with a necrotic process. In panel (**d**), relative quantification is highlighted. Three independent experiments were carried out and a representative experiment is displayed. ** denotes *p* < 0.01 vs. Q1 of the control; *** denotes *p* < 0.001 vs. Q1 of the control; °° denotes *p* < 0.01 vs. Q3 of the control; °°° denotes *p* < 0.001 vs. Q3 of the control. Analysis of variance (ANOVA) was followed by a Tukey–Kramer comparison test.

**Table 1 plants-12-02126-t001:** Measurement of phytochemical compound concentration.

Extract	Polyphenols (mg Naringin/g Extract)	Flavonoids (mg EQ/g Extract)
BF	48.8 ± 5 mg	6.21 ± 2.9 mg

## Data Availability

Not available.

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
