# Peer review of "Protective Role of an Extract Waste Product from Citrus bergamia in an In Vitro Model of Neurodegeneration"

_plants, 2023, doi:10.3390/plants12112126_

Round 1
Reviewer 1 Report
This study shows protective effects of an extract waste product from Citrus bergamia in vitro in a model of neurotoxicity induced by treatment with amyloid beta protein. It demonstrate an interesting use of a waste product. I only have a few minor comments:
1. Was the cytotoxicity checked at later time points as well?
2. In figure 5b, mention the name of the name of the protein probed in immunofluorescence i.e beta amyloid
Author Response
Dear Reviewer,
thanks for your questions that give me the opportunity to better describe this experimental work. All corrections to your comments are highlighted in green.
Jessica Maiuolo

Reviewer 2 Report
In this article the authors illustrate the protective action of the residue of Citrus bergamia extracts against neurodegenerative mechanisms using an in vitro model of neuron cultures.
Dear authors, the article is well written, well presented and full of data and information.
Anyway, some considerations:
- In the abstract it would be necessary to include information on the results obtained.
- Materials and methods: better order the part relating to the material and that relating to the HPLC analyses, separating the paragraphs
- HPLC analysis: specify how the peaks were identified, based on which standards, at which concentration. In a work published by Baron et al from 2021 polyphenols were identified from the same extracts but using MS; it is not easy to separate some PPs in C18 such as Brutieridin and Melitidin for example, as also shown by Guagliandolo et al (2019).
- In Figure 3, specify the positive control.
- Regarding the neuroprotective effect: There is no result regarding the effect of the extract on another cell lines. These data should be included in order to find the selectivity index. There are also few information on which basis you have selected these cell lines for evaluating the neuroprotective effect, although you mentioned that there are few papers showing the effect of some specific flavonoid on prevention mechanisms of neurodegenerative diseases. So, Why you did not use a pool of standards like Brutieridin Melitidin Naringin and other you found in the BF to show more clearly the role of these compounds, treating the cell coltures together with the raw extract?
- Please try to explain better why the concentration of 10 ug/ml of extract is the best concentration to observe a good effect …
english language is fine. please check some formatting and digiting error
Author Response
Dear Reviewer,
thank you for your helpful and kind advice that give me the opportunity to better describe this experimental work. All corrections to your comments are highlighted in green.
Jessica Maiuolo
